# Rapid and Easy High-Molecular-Weight Glutenin Subunit Identification System by Lab-on-a-Chip in Wheat (*Triticum aestivum* L.)

**DOI:** 10.3390/plants9111517

**Published:** 2020-11-09

**Authors:** Dongjin Shin, Jin-Kyung Cha, So-Myeong Lee, Nkulu Rolly Kabange, Jong-Hee Lee

**Affiliations:** Department of Southern Area Crop Science, National Institute of Crop Science, RDA, Miryang 50424, Korea; jknzz5@korea.kr (J.-K.C.); olivetti90@korea.kr (S.-M.L.); rollykabange@korea.kr (N.R.K.); ccriljh@korea.kr (J.-H.L.)

**Keywords:** HMW-GS, Lab-on-a-chip, electropherogram, wheat

## Abstract

Lab-on-a-chip technology is an emerging and convenient system to easily and quickly separate proteins of high molecular weight. The current study established a high-molecular-weight glutenin subunit (HMW-GS) identification system using Lab-on-a-chip for three, six, and three of the allelic variations at the *Glu-A1*, *Glu-B1*, and *Glu-D1* loci, respectively, which are commonly used in wheat breeding programs. The molecular weight of 1Ax1 and 1Ax2* encoded by *Glu-A1* locus were of 200 kDa and 192 kDa and positioned below 1Dx subunits. The HMW-GS encoded by *Glu-B1* locus were electrophoresed in the following order below 1Ax1 and 1Ax2*: 1Bx13 ≥ 1Bx7 = 1Bx7^OE^ > 1Bx17 > 1By16 > 1By8 = 1By18 > 1By9. 1Dx2 and Dx5 showed around 4-kDa difference in their molecular weights, with 1Dy10 and 1Dy12 having 11-kDa difference, and were clearly differentiated on Lab-on-a-chip. Additionally, some of the HMW-GS, including 1By8, 1By18, and 1Dy10, having different theoretical molecular weights showed similar electrophoretic mobility patterns on Lab-on-a-chip. The relative protein amount of 1Bx7^OE^ was two-fold higher than that of 1Bx7 or 1Dx5 and, therefore, translated a significant increase in the protein amount in 1Bx7^OE^. Similarly, the relative protein amounts of 8 & 10 and 10 & 18 were higher than each subunit taken alone. Therefore, this study suggests the established HMW-GS identification system using Lab-on-a-chip as a reliable approach for evaluating HMW-GS for wheat breeding programs.

## 1. Introduction

Wheat (*Triticum aestivum* L.) is an important staple food crop, which provides substantial amounts of various components, such as proteins and vitamins, that are essential for human consumption and health and the industry. In addition to being an important source of energy, wheat serves as an ingredient for diverse foods due to the presence of the seed storage protein gluten [1], which is built up of subunits and imparts elasticity to a dough [2]. 

Gluten that affects the end-use quality of common wheat, also known as bread wheat, consists of glutenins and gliadins. The glutenins are protein aggregates divided into high-molecular-weight (HMW-GS, 70~140 kDa) and low-molecular-weight (LMW-GS, 30~50 kDa) subunits [3]. HMW-GS represent approximately 10% of the total seed storage proteins and critically determine the strength and elasticity of dough with LMW-GS [4]. 

An x-type and a y-type subunits of HMW-GS encoded in each *Glu-1* locus are located on the long arms of chromosome 1 on the A, B, and D genomes of bread wheat [5]. These genes on the *Glu-1* loci are tightly linked, considering that the physical distance between an x-type and a y-type subunits ranges between 50–180 kb [6]. Six proteins of HMW-GS are translated from three *Glu-1* loci. However, some of the HMW-GS proteins are not translated due to gene silencing via elusive mechanisms, and the *Glu1-Ay* allele is mainly silenced in bread wheat [7]. Therefore, three to five glutenin subunits are found in most bread wheats. The natural allelic diversity for each subunit on the *Glu-1* loci was previously reported [8]. The recent development of advanced technologies and analysis equipment allowed the discovery of new allelic forms on the *Glu-1* loci from diverse landraces [9]. Reports indicate that endosperm storage proteins encoding genes in wheat are located at nine complex loci on six different chromosomes. Additionally, *Glu-A1*, *Glu-B1*, and *Glu-D1* possess genes for the high-molecular-weight subunits of glutenins, which are close to the centromeres on the long arms of homologous group-1 chromosomes 1A, 1B, and 1D, respectively [10]. The A subunits, also known as high-molecular-weight (HMW) subunits, are said to be well-resolved by sodium dodecyl sulphate-polyacrylamide gel electrophoresis (SDS-PAGE) [11]. To date, 22 alleles for *Glu-A1*, 52 for *Glu-B1,* and 36 for *Glu-D1* have been identified [1]. However, some of the alleles at the *Glu-1* loci are mainly found in breeding varieties [12]. For instance, *Glu-A1a* (46.8%) and *Glu-A1c* (50.9%) alleles at the *Glu-A1* locus are mainly found in current Chinese commercial wheat cultivars, and two of alleles, *Glu-B1b* (30.5%) and *Glu-B1c* (45.3%), are abundant at the *Glu-B1* locus. In addition, the frequency of the *Glu-D1a* and *Glu-D1d* alleles at the *Glu-D1* locus are 57.6% and 41.9%, respectively [13]. Moreover, five alleles, *Glu-B1a*, *Glu-B1b*, *Glu-B1c*, *Glu-B1g*, and *Glu-B1i*, at the *Glu-B1* locus are present in Australian wheat cultivars, but the major allele (*Glu-B1i*) frequency was 35.9% [14]. The *Glu-D1d* allele at *Glu-D1* was widely used for bread wheat across the globe [15]. 

The *Glu-A1b* encoding 1Ax2* at *Glu-A1* is commonly associated with good dough strength and better bread-making quality [16]. *Glu-A1a* encoding 1Ax1 at *Glu-A1* has been reported to have a high gluten index and a long dough development time compared to *Glu-A1b* [17]. Several alleles at the *Glu-B1* locus have been previously used. Among them, *Glu-B1f* (encoding the subunit pair 1Bx13 + 1By16) and *Glu-B1i* (encoding the subunit pair 1Bx17 + 1By18) had a positive effect on the dough rheological properties and bread-making quality, observed through an increase in the gluten content and loaf volume [18]. Previous reports indicated that the transcripts of 1Bx7^OE^ encoded by *Glu-B1al* were overexpressed by an insertion of 43 bp in the promoter region and/or gene duplication [19,20]. *Glu-B1al* was shown to contribute to improving the rheological properties, such as dough strength in hard wheat [21,22]. Similarly, a study by Wang and colleagues [23] revealed that *Glu-D1d* (encoding the subunit pair 1Dx5 + 1Dy10) at the *Glu-D1* locus had a positive effect on the rheological properties and dough quality. Additionally, according to Zhao et al. [22], a combination of *Glu-B1al* and *Glu-D1d* improved significantly the bread-making quality, such as dough-mixing time and strength. Some of the *Glu-1* alleles that include *Glu-A1c* (encoding the 1Ax null), *Glu-B1d* (encoding the subunit pair 1Bx6 + 1By8), and *Glu-D1a* (encoding the subunit pair 1Dx2 + 1Dy12) have been associated with poor baking quality [24,25]. However, the effects of rare alleles that are not used in current breeding cultivars have not been investigated yet. 

For several decades, the numbering system based on the mobility of the HMW-GS on sodium dodecyl sulfate-polyacrylamide gel electrophoresis (SDS-PAGE) has been widely employed to identify the allelic variation of HMW-GS [26]. Reverse-phase high-performance liquid chromatography (RP-HPLC) was developed and used to discriminate the allelic variation of HMW-GS [5,27]. In addition, several other methods, such as high-performance capillary electrophoresis (HPCE), matrix-assisted laser desorption/ionization time-of-flight mass spectrometry (MALDI-TOF-MS), and Lab-on-a-chip capillary electrophoresis (Lab-on-a-chip) were applied to identify the composition of HMW-GS [28,29,30,31]. Of these methods, MALDI-TOF-MS is said to be appropriate for detecting new allelic variations of HMW-GS [29]. However, SDS-PAGE, RP-HPLC, and MALDI-TOF-MS analysis are time-consuming and require a certain number of processes [32]. In contrast, Lab-on-a-chip has been recognized for being fast and easy to identify and quantify HMW-GS. However, this method is not properly used due to the fact that Lab-on-a-chip has not been completely established, specifically for HMW-GS identification [31]. Interestingly, the protein weight of HMW-GS on Lab-on-a-chip was shown to be higher than that observed on SDS-PAGE, and the electrophoresis pattern of HMW-GS on Lab-on-a-chip was different from SDS-PAGE. Useful molecular markers, such as Kompetitive Allele-Specific PCR (KASP) and single-nucleotide polymorphism (SNP) markers, were developed to identify the allelic variation for the *Glu-1* loci [32,33]. These molecular markers are useful for the high-throughput identification of allelic variations of genetic resources and the screening of breeding lines for quality improvement [33]. However, they do not have the ability to examine all allelic variations at the same time. Therefore, owing to the above, there is a sustaining need for the development of allelic specific markers for HMW-GS identification [33].

Reports indicate that the annual wheat flour consumption per capita in Korea is about 32 kg, the second largest after rice. However, the self-sufficiency rate is less than 2%. Developing new cultivars with a good quality for making noodles and bread are required to improve the self-sufficiency rate in Korea. In this study, we established a numbering system of HMW-GS for Lab-on-a-chip to easily identify the HMW-GS in a relatively short time. The newly developed numbering system was verified to be effective and reliable and could be applied in breeding programs. 

## 2. Results

### 2.1. Identification of 1Ax1 and 1Ax2* at the Glu-A1 Locus by Lab-on-a-Chip

The current study established a numbering system for HMW-GS identification with Lab-on-a-chip technology parallel to the widely used SDS-PAGE. We included wheat cultivars with known HMW-GS compositions using standard cultivars (Table 1). These cultivars covered three, six, and three of the allelic variations at the *Glu-A1*, *Glu-B1*, and *Glu-D1* loci, respectively, which are commonly used in wheat breeding programs. When comparing the electrophoresis patterns of HMW-GS by Lab-on-a-chip, 1Dx2.2 was by default set as the upper marker because of its large size compared to the systematic upper marker (240 kDa). Therefore, the upper marker in the standard cultivars harboring 1Dx2.2, such as Uri and Seodun, was adjusted before analyzing the HMW-GS composition (Figure 1). The recorded molecular weights of HMW-GS on Lab-on-a-chip were above 120 kDa [31]. 

To validate the numbering system for the *Glu-A1* locus on the Lab-on-a-chip system, we initially compared the observed banding patterns in Jokyung, Keumgang, Uri, and Petrel. Wheat varieties Jokyung and Keumgang harbored the same HMW-GS except at the *Glu-1* locus, while Uri and Petrel carried the 1Ax null subunit on the *Glu-A1* locus. In addition, the gel image of Jokyung and Keumgang, which carry the 1Ax1 and 1Ax2* subunits, respectively, showed that only two protein bands of about 200.6 kDa and 192.4 kDa marked the difference between Jokyung and Keumgang (Figure 1 and Appendix A). However, these protein bands could not be found in Uri and Petrel harboring the 1Ax null subunit. Therefore, these data revealed that the observed protein bands of 200.6 kDa and 192.4 kDa on Lab-on-a-chip are Glu-1Ax1 and Glu-1Ax2*, respectively.

### 2.2. Electrophoresis Patterns of HMW-GS Encoded by the Glu-B1 Locus

In order to validate the electrophoresis banding patterns of HMW-GS encoded by the *Glu-B1* locus on Lab-on-a-chip, we used six standard wheat cultivars covering six alleles. To determine the molecular weight of 1Bx7 on Lab-on-a-chip, we first examined the HMW-GS of Petrel, in which the 1Bx7 subunit was only translated from the *Glu-B1* locus. The protein band size of 177.8 kDa in Petrel was identified as 1Bx7 on Lab-on-a-chip, which was similar with the molecular weight of the 1Bx7 (177 kDa) subunit reported earlier [36]. Additionally, similar protein bands were identified in wheat varieties Seodun and Anbaek, carrying the 1Bx7 subunit. Meanwhile, Seodun and Anbaek harbored different HMW-GS at the *Glu-1B* locus, 1Bx7 + 1By8 and 1Bx7 + 1By9, respectively. When the electrophoresis patterns of HMW-GS in Seodun and Anbaek were compared to one another, 133.9 kDa and 129.9 kDa of the protein bands were detected (Figure 1). A protein band of 133.9 kDa was examined in varieties carrying the 1By8 subunit, such as Jokyung and Uri. Owing to the above, these results suggest that the detected protein band size of 177.8 kDa in Petrel indicates 1Bx7, while the one of 133.9 kDa in Seodun and 129.9 kDa in Anbaek are 1By8 and 1By9, respectively (Figure 1).

The wheat variety Joeun harboring 1Bx13 + 1By16 at the *Glu-B1* locus was used to establish the electrophoresis pattern of 1Bx13 + 1By16. The 171.7 kDa and 141.4 kDa protein bands in Joeun were clearly different compared with Seodun and Anbaek, carrying 1Bx7 + 1By8 and 1Bx7 + 1By9, respectively (Figure 1 and Appendix A). The molecular weight of 1Bx13 was higher than that of the 1Bx16 protein observed on SDS-PAGE. Thus, the upper and lower bands in Joeun are considered as 1Bx13 and 1By16, respectively. The molecular weight of 1Bx13 is about 1 kDa larger than that of 1Bx7. The 1Bx13 protein band was located a little more upper than 1Bx7 on Lab-on-a-chip (Figure 1). Meanwhile, 1By16 was positioned above 1By8 with 5-kDa difference on Lab-on-a-chip and was clearly distinguished from other HMW-GS.

To determine the position of 1Bx17 + 1By18 and 1Bx7^OE^ + 1By8 on Lab-on-a-chip, we used Joongmo2008 and Vesna possessing 1Bx17 + 1By18 and 1Bx7^OE^ + 1By8 at the *Glu-B1* locus, respectively. The electrophoresis pattern of Joongmo2008 carrying both 1Bx17 + 1By18 was compared with Petrel, which harbors 1Bx7. The molecular weight of the 1Bx17 protein on Lab-on-a-chip was 159.6 kDa and was clearly differentiated from other subunits (Figure 1). However, we could not distinguish the 1By18 and 1Dy10 protein bands on Lab-on-a-chip. Nevertheless, the 1Bx17 + 1By18 subunit could be identified using the electrophoresis position of the 1Bx17 subunit from other HMW-GS. In addition, the gel-like image of Vesna harboring 1Bx7^OE^ + 1By8 was investigated with the purpose of validating the electrophoresis position of 1Bx7^OE^ + 1By8 (Figure 1). Interestingly, the electrophoresis pattern of 1Bx7^OE^ + 1By8 on Vesna was the same with Jokyung harboring 1Bx7 + 1By8, considering that 1Bx7^OE^ encoded by *Glu-B1al* are overexpressed by the insertion of 43 bp in the promoter and/or gene duplication of 1Bx7 [19,20]. We could not separate 1Bx7^OE^ + 1By8 from 1Bx7 + 1By8 with the gel-like image of Lab-on-a-chip.

### 2.3. Identification of the 1Dx5 + 1Dy10 Subunit from Other Subunits Encoded from the Glu-D1 Locus

To validate the electrophoresis pattern of HMW-GS encoded by the *Glu-D1* locus, we compared the three kinds of alleles in the *Glu-D1* locus: 1Dx2 + 1Dy12, 1Dx5 + 1Dy10, and 1Dx2.2 + 1Dy12. The molecular weights of 1Dx2 in Anbaek, 1Dx5 in Petrel, and 1Dx2.2 in Seodun were 229.0 kDa, 224.0 kDa, and 281.7 kDa, respectively (Figure 1 and Appendix A). We mentioned earlier that 1Dx2.2 was located above the systemic upper marker (240 kDa). The molecular weight of 1Dy10 in Petrel and 1Dy12 in Seodun were 133.7 kDa and 123.3 kDa, respectively. The data showed that 1Dx subunits encoded by the *Glu-D1* locus were positioned above the 1Ax1 and 1Ax2*-encoded *Glu-A1* locus, even though the theoretical molecular weight of the 1Dx subunits were smaller than the 1Ax subunits. The electrophoresis of 1Dy12 on Lab-on-a-chip was faster than the other HMW-GS and was placed at the bottom (Figure 1). When the band sizes of these 1Dx were compared with one another, 1Dx5 was discriminated from 1Dx2. However, it was a bit difficult to distinguish 1Dx2 and 1Dx5, because the molecular weights of these proteins by Lab-on-a-chip were slightly different.

### 2.4. Discrimination of 8 & 10, 10 & 18, and 7^OE^ Subunit by Analyzing the Electropherogram

HMW-GS were clearly separated by SDS-PAGE. However, some of the HMW-GS were electrophoresed, including the one with the same molecular weight on Lab-on-a-chip: 1By8, 1By18, and 1Dy10. Additionally, the 1Bx7^OE^ subunit showed the same molecular weight with the 1Bx7 subunit. Therefore, it is difficult to distinguish subunits clearly with a gel-like image or molecular weights of Lab-on-a-chip (Figure 1, Appendix A). To distinguish the 1By8 + 1Dy10 (here referred to as 8 & 10) subunit from 1By8 and 1Dy10, we used three wheat varieties as the background to perform an electrophoresis by Lab-on-a-chip. The results showed that the gel-like image and electropherogram of the 1By8, 1Dy10, and 8 & 10 subunits had similar protein band positions based on their molecular weights on Lab-on-a-chip (Figure 2 and Appendix A). However, the protein amount indicated by the peak height on the electropherogram for 8 & 10 was about two-fold higher than that of 1By8 and 1Dy10 taken alone, despite the fact that the band peak height of 1Bx7 in Hanbaek carrying 8 & 10 was slightly lower than the one observed in Saekeumgang and Chapingo (Figure 2). Interestingly, the recorded protein amount indicated by the peak height and relative protein quantity of 8 & 10 was about two-fold higher than that of 1Bx7 (Table 2).

Additionally, the discrimination of the 1By18 + 1Dy10 (referred to as 10&18) and 1Dy10 protein bands indicated in the electropherogram of Chapingo and Garnet, which harbor 1Dy10 and 10 & 18, respectively, revealed that the protein amount of 10 & 18 was about two-fold higher than that of 1Dy10 alone (Table 3). Meanwhile, the band peak height of 1Ax2* and 1Dx5 showed similar patterns (Figure 3). Furthermore, 8 & 10 and 10 & 18 exhibited similar protein amounts (Figure 3). Thus, it is believed that, when 8 & 10 and 10 & 18 are electrophoresed together on Lab-on-a-chip, these subunits could be distinguished by analyzing the relative protein amounts on an electropherogram (Figure 3 and Table 3).

Besides, the 1Bx7^OE^ + 1By8 subunit was identified on Lab-on-a-chip in nine wheat varieties that carry similar HMW-GS, without 1Bx7 or 1Bx7^OE^. The results indicated that all the selected varieties exhibited a similar electrophoresis pattern on the gel-like image, but 1Bx7^OE^ showed a high band density compared to that of the other HMW-GS (Figure 4). The analysis of the electropherogram of the nine wheat varieties showed that 1Bx7^OE^ had peak heights of about two-fold higher than those of 1Dx5 or 8&10 (Appendix A). Despite the fact that 1Bx7^OE^ and 1Bx7 showed similar protein sizes on Lab-on-a-chip, 1Bx7^OE^ could be distinguished from 1Bx7 by its protein amount (thick band) on the gel-like image and relative protein amount on the electropherogram in Lab-on-a-chip.

### 2.5. HMW-GS Composition Identification of Genetic Resources by Lab-on-a-Chip

To examine whether the established numbering system with Lab-on-a-chip could be effectively used for HMW-GS identification in a wheat breeding program, we tested the HMW-GS of 121 varieties by both the gel-like image and electropherogram of Lab-on-a-chip. The data showed that 1Ax1, 1Ax2*, and 1Ax null at the *Glu-A1* locus were found in 51, 56, and 14 verities with their respective molecular weights on Lab-on-a-chip of 201.5 kDa and 191.8 kDa (Figure 5). Of all the HMW-GS at the *Glu-B1* locus, the 1Bx7 + 1By9 or 1Bx17 + 1By18 subunits were detected at a high frequency compared to the composition of the other subunits. In essence, 34 varieties carried the 1Bx7 + 1By9 subunit, while the 1Bx13 + 1By16 and 1Bx7^OE^ + 1By8 subunits were found in 10 and 18 varieties, respectively. However, no variety carried an independent 1Bx7 subunit. The average molecular weight of these subunits encoded by the *Glu-B1* locus ranged from 125.9–168.9 kDa on Lab-on-a-chip (Appendix A). Whereas 1Bx7 and 1Bx7^OE^ exhibited an average molecular weight of 166.6 kDa and 166.8 kDa, and 1Bx13 and 1Bx17 showed 168.9 kDa and 153.9 kDa, respectively. Subunits 1By8, 1By9, 1By16, and 1By18 had 133.7 kDa, 125.9 kDa, 139.3 kDa, and 133.9 kDa, respectively. Moreover, 105, 3, and 13 of varieties harbored the 1Dx5 + 1Dy10, 1Dx2 + 1Dy12, and 1Dx2.2 + 1Dy12 encoded by the *Glu-D1* locus. The average molecular weights of 1Dx5, 1Dx2, and 1Dx2.2 on Lab-on-a-chip were 210.8 kDa, 218.0 kDa, and 277.3 kDa, while the ones of 1Dy10 and 1Dy12 were 133.2 kDa and 122.4 kDa, respectively (Figure 5).

## 3. Discussion

Recent decades have been marked by significant progress in the genetic characterization of gluten proteins in wheat using improved procedures or methods of protein fractionation and the higher availability of genetic resource stocks. 

The Lab-on-a-chip electrophoresis system is a reliable and efficient technology for fast and easy protein separation and quantification. In addition, Lab-on-a-Chip equipment offers the unique comparative advantage of being deployable beyond the laboratory compared to conventional methods of protein analysis. However, it still needs to be established for the HMW-GS identification for wheat breeding. Earlier, a study conducted by Rhazi et al. [37] investigated the separation and quantification of HMW-GS in wheat using a high-throughput microchip capillary electrophoresis-sodium dodecyl sulfate (microchip CE) platform, the LabChip 90 system. Their study proposed that the microchip CE analysis could provide a comparable resolution and sensitivity to conventional RP-HPLC for the identification of HMW-GS but faster compared to the latter. In the same way, Uthayakumaran and his colleagues [38,39] proposed a similar separation and identification system of HWM-GS with a longer sample processing time using an Agilent 2100 Bioanalyzer with a Protein 200 + LabChip.

In addition, other studies have established a numbering system for HMW-GS identification by SDS-PAGE, based on the molecular weight of HMW-GS [5]. The HMW-GS were named by electrophoresis mobility on SDS-PAGE, and this system has been used for HMW-GS identification for years [2]. However, the HMW-GS identification system by SDS-PAGE was shown to be time-consuming and required a large gel system to clearly separate HMW-GS. Many research groups have tried to apply the Lab-on-a-chip system for HMW-GS identification, due to the fast and effective protein separation and quantification that offers the Lab-on-a-chip electrophoresis system [31,40]. However, the latter system is not commonly used for HMW-GS identification due to particular reasons [31]. On the one hand, the molecular weight of HMW-GS on Lab-on-a-chip was shown to be higher than that observed on SDS-PAGE [31]. 

In the current study, we distinguished and validated the exact molecular weights and quantified each HMW-GS on Lab-on-a-chip using standard wheat varieties and other genetic resources. On the one hand, the theoretical molecular weight of HMW-GS ranged between 70–150 kDa, but the molecular weights investigated in standard varieties and genetic resources on Lab-on-a-chip ranged between 120–280 kDa (Figure 5). On the other hand, the electrophoresis patterns of 1Ax1 and 1Ax2* on the Lab-on-a-chip system were found to be different from the one on SDS-PAGE. The 1Ax1 and 1Ax2* subunits were detected above the 1Dx2 and 1Dx5 subunits in SDS-PAGE, considering its theoretical molecular weight. However, these subunits were positioned below the 1Dx2 and 1Dx5 subunits on Lab-on-a-chip (Figure 1) [2,40]. It has been reported that Tris-acetate acrylamide gel is ideal for separating large-molecular-weight proteins. In our previous study, when HMW-GS proteins were electrophoresed in 3–8% of gradient Tris-acetate acrylamide gel on SDS-PAGE, the 1Ax1 and 1Ax2* proteins were detected at a lower position than Dx2 and Dx5 in this gel condition like Lab-on-a-chip [35]. This phenomenon was also observed when acid polyacrylamide gel (A-PAGE) was applied for HMW-GS separation [30]. In addition, the 1Dy10 subunit was clearly separated from the 1By8 and 1By18 subunits on SDS-PAGE. The application of another Lab-on-a-chip system, the Experion Pro 260 assay kit (Bio-Rad, Hercules, CA, USA), revealed that the electrophoresis patterns of HMW-GS were different from those observed with a Protein 230 assay kit (Agilent Technologies, Palo Alto, CA, USA). The 1Dy10 protein was discriminated from 1Bx8, but 1Bx8 was found to overlap with 1Dy12 in the Experion Pro 260 assay kit [41]. We did not find the main reason for the 1Dy10 subunit to be slowly electrophoresed on Lab-on-a-chip with the Protein 230 assay kit. However, it is thought that it could be due to the buffer and gel system of the Protein 230 assay kit for Lab-on-a-chip. The 240-kDa protein considered as a systemic upper marker is used to determine the protein molecular weight on Lab-on-a-chip. Any protein that could be detected in the sample having a larger size than the upper marker was, by default, set as the systemic upper marker. The molecular weight of 1Dx2.2 on Lab-on-a-chip was higher than the systemic upper marker, which led to the adjustment of the systemic upper marker position prior to analyzing the HMW-GS composition (Figure 1).

Jang et al. [5] evaluated the HMW-GS composition wheat varieties, and they used 16 standard wheat varieties for HMW-GS identification and 38 Korean wheat cultivars by RP-HPLC and SDS-PAGE [5]. In our case, to establish the HMW-GS identification in the wheat breeding program on Lab-on-a-chip, we used nine varieties of which the HMW-GS were identified earlier [5]. We then screened Vesna carrying 1Bx7^OE^ by the Kompetitive allele-specific PCR (KASP) assay with the Bx7^OE^_866_SNP marker and later used it for 1Bx7^OE^ subunit identification [35]. These varieties covered three, six, and three allelic variations at the *Glu-A1*, *Glu-B1*, and *Glu-D1* loci, respectively (Figure 1). They did not cover all allelic variations of the *Glu-1* loci, but we thought that the allelic variations used in this study were enough to be applied in wheat breeding programs. In the present study, four varieties were used to specify the molecular weight of 1Ax1, 1Ax2*, and 1Ax null encoded by the *Glu-A1* locus. We first found 1Ax1 of Jokyung and 1Ax2* of Keumgang by excluding the same positional proteins after comparing with Uri and Petrel harboring the 1Ax null. The molecular weights of 1Ax1 and 1Ax2* were lower than 1Dx2 and 1Dx5 on Lab-on-a-chip (Figure 1) [31,41]. Previously, diverse alleles were reported in the *Glu-B1* locus. We determined the molecular weight and relative quantity of six alleles in the *Glu-B1* locus on Lab-on-a-chip [42]. The molecular weight of HMW-GS encoded by the *Glu-Bl* locus was shown in the following order: 1Bx13 ≥ 1Bx7 = 1Bx7^OE^ > 1Bx17 > 1By16 > 1By8 = 1By18 > 1By9. The molecular weight of 1Bx13 was slightly higher than that of 1Bx7 and 1Bx7^OE^, and the electrophoresis mobility of 1By8 was similar with 1By18 on Lab-on-a-chip.

However, x-type and y-type HMW-GS moved like one gene, because the physical distance between these genes was about 50~180 kb [7]. So, the 1Bx7 + 1By8, 1Bx7 + 1By9, 1Bx13 + 1By16, and 1Bx17 + 1By18 compositions were discriminated using the molecular weights of 1By8, 1By9, 1By16, and 1Bx17, respectively (Figure 5). Additionally, the composition of 1Bx7 + 1By8 was effectively separated from other HMW-GS by investigating the relative protein quantity on an electropherogram of Lab-on-a-chip (Appendix A). 

In the case of HMW-GS on the *Glu-D1* locus, 1Dx2.2 + 1Dy12 was clearly observed, and this was facilitated by the position of 1Dx2.2 above the systematic upper marker. The molecular weight of 1Dx2 was 7 kDa higher than 1Dx5, though, in similar cases, it was not always easy to distinguish 1Dx2 from 1Dx5 on Lab-on-a-chip. However, 1Dy12 was faster electrophoresed than 1Dy10. Thus, 1Dx2 + 1Dy12 and 1Dx5 + 1Dy10 were identified by examining 1Dy12 on a gel-like image of Lab-on-a-chip (Figure 1).

Subunits 1By8 and 1Dy10 were clearly differentiated by Lab-on-a-chip with the Experion Pro 260 assay kit [41]. Three of the HMW-GS, 1By8, 1By18, and 1Dy10, were electrophoresed as having similar molecular weights on the Lab-on-a-chip with the Protein 230 assay kit (Figure 2 and Figure 3). It is then believed that the difference between the Experion Pro 260 assay kit and Protein 230 assay kit of Agilent could be partly explained by different buffer systems for Lab-on-a-chip. Despite the existing differences in the protein electrophoresis mobility of HMW-GS between manufacturers, the 8 & 10 and 10 & 18 bands could be distinguished from 1By8, 1By18, and 1Dy10 alone by analyzing the relative protein quantity of the electropherogram (Figure 2 and Figure 3). Additionally, no reports could be found elaborating on the electropherogram analysis to identify HMW-GS currently, but the electropherogram analysis was found to be an important way for clear HMW-GS identification, such as 8 & 10.

Besides, the reported molecular markers were useful for high-throughput HMW-GS identification, and KASP assay makers for HMW-GS identification were recently developed for rapid genotyping [33,35]. However, it still requires independent experiments for each HMW-GS. Lab-on-a-chip takes about 25 min to process 10 samples for HMW-GS identification, and approximately 120 samples could be tested in a day [31]. All HMW-GS could be identified and quantified at the same time. Therefore, this study established an HMW-GS identification system for Lab-on-a-chip, with standard varieties covering diverse HMW-GS. The Lab-on-a-chip system is relatively easier and faster than SDS-PAGE and RP-HPLC. Nevertheless, downstream studies are required to validate for minor alleles of HMW-GS. Owing to the above, this study suggests that the Lab-on-a-chip system could be served as a reliable and effective technology to identify and quantify the HMW-GS for the wheat breeding program.

## 4. Materials and Methods 

### 4.1. HMW-GS Composition Identification of Genetic Resources by Lab-on-a-Chip

Nine wheat varieties covering diverse HMW-GS were used to develop the numbering system for the HMW-GS identification by Lab-on-a-chip (Table 3). These varieties included three alleles at the *Glu-A1* locus, six alleles at the *Glu-B1* locus, and three alleles at the *Glu-D1* locus. Three, four, and eight varieties were used to evaluate the protein amounts of 8 & 10, 10 & 18, and 7^OE^ subunits, respectively, on an electropherogram. Another set of 121 wheat varieties were obtained from the National Agrobiodiversity Center, National Institute of Agricultural Science, Rural Development Administration (RDA), Republic of Korea. These varieties were used to examine and verify the efficiency and reliability of the newly developed numbering system for HMW-GS identification. 

The glutenin was extracted from wheat flour following the procedure reported by Van Den Broeck et al. [43]. Briefly, a mixture of approximately 100 mg of flour and 1 mL of 50% (*v*/*v*) propanol was incubated for 30 min at 65 °C, followed by centrifugation at 10,000× *g* for 5 min, and the supernatant containing gliadin was discarded. Then, the precipitate was suspended in 0.7-mL 80-mM Tris-HCl (pH 8.0) containing 2% SDS and 1% (*w*/*v*) dithiothreitol (DTT) and incubated at 65 °C for 30 min. The mixture was centrifuged at 10,000× *g* for 5 min, and 0.3 mL of the buffer containing 1% DTT was added and incubated at 65 °C for 15 min. After centrifugation at 10,000× *g* for 5 min, the supernatant was collected and used for downstream analysis by Lab-on-a-chip.

The Protein 230 assay kit (Agilent Technologies, Palo Alto, CA, USA) was used for HMW-GS identification. A total of 12 μg (3 μg/μL) of extract glutenin proteins were mixed with 2 μL of denaturing solution. After heating at 95 °C for 5 min, 84 μL of deionized water was added to the sample tube. Six microliters of diluted samples were loaded on the chip. Then, the diluted samples were separated on the 2100 Bioanalyzer (Agilent Technologies, Palo Alto, CA, USA), based on gel electrophoresis principles replicated onto a chip format following the manufacturer’s instructions [44]. The upper marker (240 kDa) was adjusted before analyzing the HMW-GS for varieties harboring 1Dx2.2. The protein concentration was quantified with electropherogram on the 2100 Expert program (Agilent Technologies, Palo Alto, CA, USA). Samples were collected in triplicate for each wheat variety used in the study. The values of protein amounts obtained from different band picks were compared using the Student’s *t*-test (*p* < 0.05). 

### 4.2. Glutenin Proteins Extraction and HMW-GS Composition Identification of Genetic Resources by SDS-PAGE

Glutenin subunits were also separated by sodium dodecyl sulfate-polyacrylamide gel electrophoresis (SDS-PAGE) following the procedure described earlier [45]. Briefly, the separation gel contained 1.5-M Tris-HCl (pH 8.8) and 0.27% SDS. Gels were made of 7.5% (*w*/*v*) acrylamide and 0.2% (*w*/*v*) bis-acrylamide. The stacking gel was made of 0.25-M Tris-HCl (pH 6.8), 0.2% SDS, and 7.5% (*w*/*v*) acrylamide and 0.2% (*w*/*v*) bis-acrylamide. Wheat flour was suspended in 300-mL 0.25-M Tris-HCl buffer (pH 6.8), containing 2% (*w*/*v*) SDS, 10% (*v*/*v*) glycerol, and 5% 2-mercaptoethanol, followed by shaking for 2 h at room temperature. Then, the slurry was heated for 3 min at 95 °C, and the supernatant was subjected to SDS-PAGE.

The HMW numbering system of glutenin subunit bands and that for the allelic classification at different loci previously proposed by Payne and his colleague [11] were used in the current study. For the determination of the electrophoretic mobility of each HMW glutenin subunit by SDS-PAGE, standard wheat varieties that included the spectra of the subunits expected were used. Thus, the overall quality scores of HMW glutenin subunits for a particular variety could be obtained as the sum of the scores of each individual subunit and compared with the standard bread-making quality of the wheat varieties [46]. 

## Figures and Tables

**Figure 1 plants-09-01517-f001:**
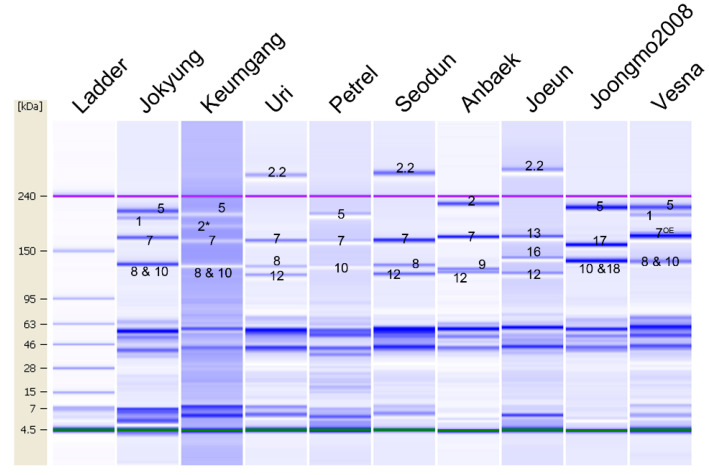
Gel-like image of the standard varieties on Lab-on-a-chip. Six microliters of diluted extracted glutenin proteins with denaturing solution, including in the protein 230 assay kit, were separated by the 2100 Bioanalyzer. The upper marker (240 kDa) was adjusted before analyzing the high-molecular-weight glutenin subunit (HMW-GS) for varieties harboring 1Dx2.2, such as Uri, with the 2100 Expert program. Then, a gel-like image was taken. The purple line indicates the systematic upper marker (240 kDa). The small numbers on the image mean each HMW-GS such as 5 is 1Dx5 and 1 is 1Ax1.

**Figure 2 plants-09-01517-f002:**
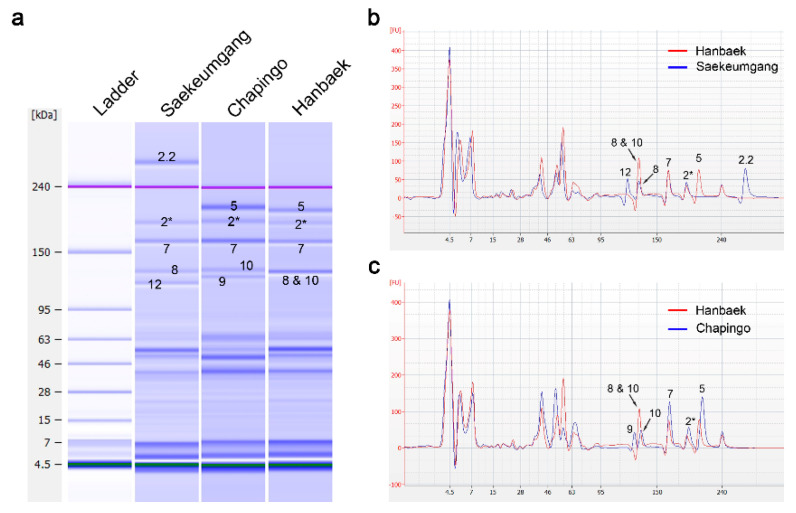
Identification of the 8 & 10 subunits on Lab-on-a-chip. (**a**) Gel-like image of HMW-GS. (**b**) Comparison of the protein quantities between 1By8 and 8 & 10 by electropherogram. (**c**) Comparison of the protein quantities between 1Dy10 and 8 & 10 by electropherogram. Six micrograms of diluted extracted glutenin proteins with denaturing solution were separated by the 2100 Bioanalyzer. The electropherograms were overlaid with the 2100 Expert program. The small numbers on the image refer to each HMW-GS, such as 2.2 is 1Dx2.2 and 12 is 1Dy12.

**Figure 3 plants-09-01517-f003:**
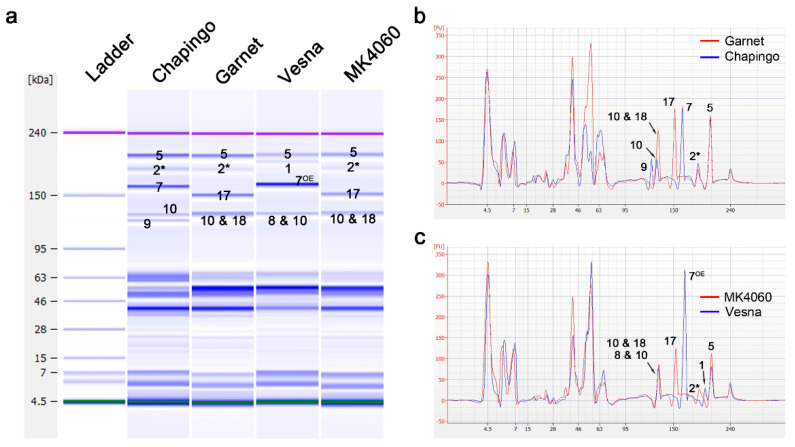
Identification of the 10 & 18 subunit on Lab-on-a-chip. (**a**) The gel-like image of HMW-GS. (**b**) Comparison of the protein quantities between 1Dy10 and 10 & 18 by electropherogram. (**c**) Comparison of the protein quantities between 8 & 10 and 10 & 18 by electropherogram. Six micrograms of diluted extracted glutenin proteins with denaturing solution were separated by the 2100 Bioanalyzer. The electropherograms were overlaid with the 2100 Expert program. The small numbers on the image mean each HMW-GS, such as 5 is 1Dx5 and 2* is 1Ax2*.

**Figure 4 plants-09-01517-f004:**
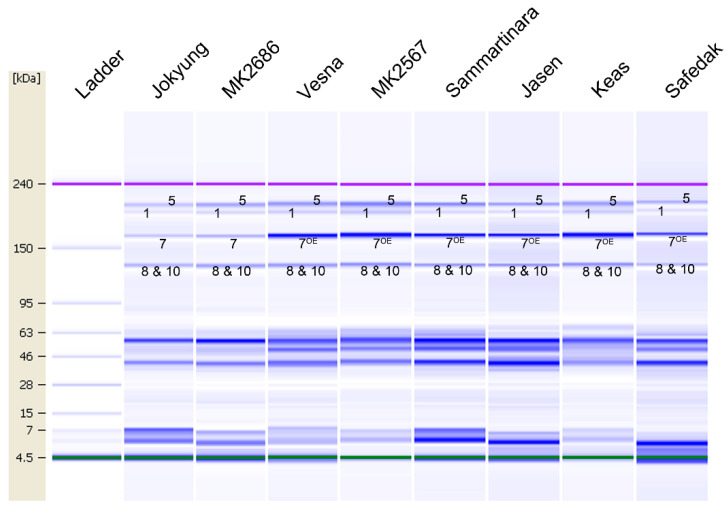
Identification of the 7^OE^ subunit on Lab-on-a-chip. Gel-like image of HMW-GS in different wheat varieties. The small numbers on the image mean each HMW-GS, such as 7 is 1Bx7 and 7^OE^ is 1Bx7^OE^.

**Figure 5 plants-09-01517-f005:**
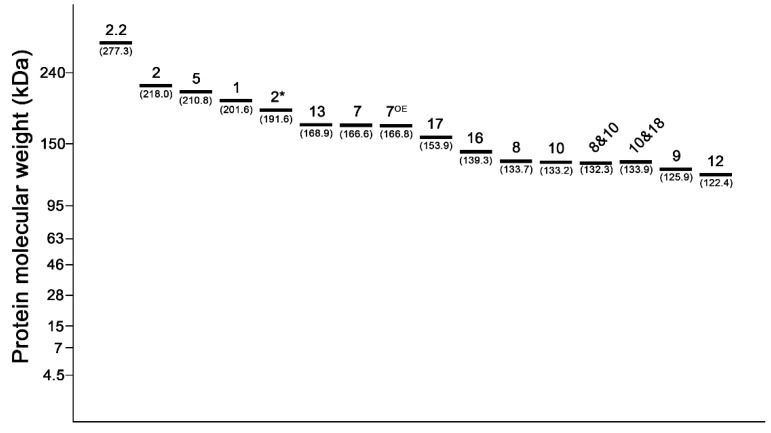
Molecular weights of HMW-GS on Lab-on-a-chip. The mean of each HMW-GS was calculated by analyzing of molecular weight of each HMW-GS from the genetic resources. The small numbers on the image mean each HMW-GS, such as 2 is 1Dx2 and 5 is 1Dx5.

**Table 1 plants-09-01517-t001:** High-molecular-weight glutenin subunit (HMW-GS) composition used as the standard varieties in this study.

Variety Name	*Glu-1* Locus Allele			SDS-PAGE Allele			Reference
	*Glu-A1*	*Glu-B1*	*Glu-D1*	*Glu-A1*	*Glu-B1*	*Glu-D1*	
Jokyung	*a*	*b*	*d*	1	7 + 8	5 + 10	Jang et al. [5]
Keumgang	*b*	*b*	*d*	2*	7 + 8	5 + 10	Jang et al. [5]
Uri	*c*	*b*	*f*	null	7 + 8	2.2 + 12	Jang et al. [5]
Petrel	*c*	*a*	*d*	null	7	5 + 10	Liu et al. [34]
Seodun	*c*	*b*	*f*	null	7 + 8	2.2 + 12	Jang et al. [5]
Anbaek	*c*	*c*	*a*	null	7 + 9	2 + 12	Jang et al. [5]
Joeun	*c*	*f*	*f*	null	13 + 16	2.2 + 12	Jang et al. [5]
Joongmo2008	*c*	*i*	*d*	null	17 + 18	5 + 10	Jang et al. [5]
Vesna	*a*	*al*	*d*	1	7^OE^ + 8	5 + 10	Shin et al. [35]

**Table 2 plants-09-01517-t002:** The relative protein quantity of HMW-GS for 8 & 10 identification by Lab-on-a-chip. The relative protein quantity of each HMW-GS was analyzed by the band peak area on an electropherogram. Three independent experiments were performed. Data are mean values (%) ± SD.

HMW-GS	Saekeumgang	Chapingo	Hanbaek
1Ax2*	11.5 ± 2.68	13.7 ± 1.26	11.0 ± 0.98
1Dx5	-	33.7 ± 3.29	34.3 ± 5.11
1Dx2.2	23.7 ± 5.08	-	-
1Bx7	31.3 ± 5.56	27.5 ± 1.85	25.8 ± 1.63
1By8	13.1 ± 3.29	-	-
1By9	-	9.9 ± 1.98	-
8 & 10	-	-	29.0 ± 5.28
1Dy10	-	15.2 ± 3.04	-
1Dy12	20.4 ± 5.49	-	-

**Table 3 plants-09-01517-t003:** Relative protein quantity of HMW-GS for 10 & 18 identification by Lab-on-a-chip. The relative protein quantity of each HMW-GS was analyzed by the band peak area on the electropherogram. Three independent experiments were performed. Values are the mean (%) ± SD.

HMW-GS	Chapingo	Garnet	Vesna	MK4060
1Ax1	-	-	9.5 ± 1.12	-
1Ax2*	13.7 ± 1.26	6.4 ± 0.34	-	9.8 ± 1.12
1Bx7	27.5 ± 1.85	-	-	-
1Bx17	-	30.2 ± 0.94	-	30.2 ± 2.62
1Bx7^OE^	-	-	45.3 ± 2.71	-
1By9	9.9 ± 1.98	-	-	-
8&10	-	-	20.4 ± 1.28	
10&18	-	29.2 ± 4.25	-	28.9 ± 2.15
1Dx5	33.7 ± 3.29	34.3 ± 3.11	24.8 ± 1.06	31.1 ± 0.96
1Dy10	15.2 ± 3.04	-	-	-

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
