# Peer review of "Rapid and Easy High-Molecular-Weight Glutenin Subunit Identification System by Lab-on-a-Chip in Wheat (Triticum aestivum L.)"

_plants, 2020, doi:10.3390/plants9111517_

Round 1
Reviewer 1 Report
The manuscript entitled “Rapid and easy high-molecular-weight glutenin subunit identification system by Lab-on-a-chip in Triticum aestivum” deals with a very important and topical subject. breeders and cereal scientists need a fast and efficient method for the identification and quantification of glutenin subunits. The study is interesting but it does not address the learned studies carried out previously (see below). In order to be innovative and to stand out from these three studies, I suggest to the authors to redo an argumentative discussion based on the comparison of their studies with these three studies. Also, a new system has been developed recently by PerkinElmer with an improved kit for high throughput analysis! Up to know this is the alone system are able to realize identification and quantitation of glutenin subunits, in particular for screening wheat quality.
I'm surprised I didn't see these three articles cited in yours!!?
Rhazi et al., 2009: High throughput microchip-based separation and quantitation of high-molecular-weight glutenin subunits. Journal of Cereal Science 49. 272–277
Uthayakumaran et al., 2006. Rapid identification and quantitation of high-molecular-weight glutenin subunits. Journal of Cereal Science, 1–6.
Uthayakumaran, et al., 2005. On-the-spot identification of grain variety and wheat-quality type by lab-on-a-chip capillary electrophoresis. Journal of Cereal Science 41, 371–374.
Author Response
Rapid and easy high-molecular-weight glutenin subunit identification system by Lab-on-a-chip in Wheat (Triticum aestivum L.)
Manuscript ID: plants-983586R1
Point by point reply to comments of reviewer
We are thankful to the editorial team and anonymous reviewer for their time dedicated to this manuscript. We sincerely appreciate the comments, which helped us improved significantly the content and the quality of the manuscript. We would like to specify that all changes in the manuscript were highlighted green. We hope that the manuscript in the present form will be suitable for publication in the journal.
|
Reviewer 1
|
The manuscript entitled “Rapid and easy high-molecular-weight glutenin subunit identification system by Lab-on-a-chip in Triticum aestivum” deals with a very important and topical subject. breeders and cereal scientists need a fast and efficient method for the identification and quantification of glutenin subunits. The study is interesting but it does not address the learned studies carried out previously (see below). |
|
In order to be innovative and to stand out from these three studies, I suggest to the authors to redo an argumentative discussion based on the comparison of their studies with these three studies. I'm surprised I didn't see these three articles cited in yours!!? Rhazi et al., 2009: High throughput microchip-based separation and quantitation of high-molecular-weight glutenin subunits. Journal of Cereal Science 49. 272–277
Uthayakumaran et al., 2006. Rapid identification and quantitation of high-molecular-weight glutenin subunits. Journal of Cereal Science, 1–6.
Uthayakumaran, et al., 2005. On-the-spot identification of grain variety and wheat-quality type by lab-on-a-chip capillary electrophoresis. Journal of Cereal Science 41, 371–374.
|
We are thankful to the Reviewer for his constructive comments and suggestions to improve the manuscript. Therefore, we have considered the methods previously proposed for separation and quantification of high molecular weight glutenin subunits, and have expanded the discussion as shown in lines 258-271. Recent decades have been marked by significant progress in the genetic characterization of gluten proteins in wheat using improved procedures or methods of protein fractionation, and higher availability of genetic resources stocks. Lab-on-a-chip electrophoresis system is a reliable and efficient technology for fast and easy protein separation and quantification. In addition, the Lab-on-a-Chip equipment offers the unique comparative advantage of being deployable beyond the laboratory compared to conventional methods of protein analysis. However, it still needed to be established for the HMW-GS subunit identification for wheat breeding. Earlier, a study conducted by Rhazi et al. [1] investigated the separation and quantification of HMW-GS in wheat using a high throughput microchip capillary electrophoresis-sodium dodecyl sulfate (microchip CE) platform, LabChip 90 system. Their study proposed that the microchip CE analysis could provide a comparable resolution and sensitivity to conventional RP-HPLC for the identification of HMW-GS but faster compared to the latter. In the same way, Uthayakumaran and his colleagues [2,3] proposed a similar separation and identification system of HWM-GS with a longer samples’ processing time using Agilent 2100 Bioanalyzer with a Protein 200+ LabChip. |
|
Also, a new system has been developed recently by PerkinElmer with an improved kit for high throughput analysis! Up to know this is the alone system are able to realize identification and quantitation of glutenin subunits, in particular for screening wheat quality. |
We appreciate the suggestion made by the Reviewer. We have noticed that PerkinElmer proposes a variety of kit for scientific research. However, we have no particular opinion on their products. |
- Rhazi, L.; Bodard, A.L.; Fathollahi, B.; Aussenac, T. High throughput microchip-based separation and quantitation of high-molecular-weight glutenin subunits. J. Cereal Sci. 2009, 49, 272-277.
- Uthayakumaran, S.; Batey, I.L.; Wrigley, C.W. On-the-spot identification of grain variety and wheat-quality type by Lab-on-a-chip capillary electrophoresis. J. Cereal Sci. 2005, 41, 371-374.
- Uthayakumaran, S.; Listiohadi, Y.; Baratta, M.; Batey, I.L.; Wrigley, C.W. Rapid identification and quantitation of high-molecular-weight glutenin subunits. J. Cereal Sci 2006, 44, 34-39

Reviewer 2 Report
The paper “Rapid and easy high-molecular-weight glutenin subunit identification system by Lab-on-a-chip in Triticum aestivum” was focused on establishing a numbering system of HMW-GS subunits for Lab-on-a-chip to easily identify the HMW-GS in a short time. The study was targeted on 3, 6, and 3 of allelic variations at Glu-A1, Glu-B1, and Glu-D1 loci, which are commonly used in the wheat breeding program.
The research topic is relevant because it is necessary to have rapid information about the main wheat protein due to its crucial effect on dough rheology and breadmaking. Such analysis could be complementary not only for providing information about low molecular weight glutenin subunits likely, but also to reflect on the contributions of growth/storage conditions on the quality of wheat and flour.
It is expected that the application of such systems will become routine in the near future
The introduction provides sufficient background and includes all relevant references.
Line 45- reference need to be quoted
Line 87-88- the authors can list some of the molecular markers for an easier understanding
Regarding the Materials and Methods section, it needs to be reorganized in an easier-to-follow manner (e.g samples, protein extraction, Lab-on-a-chip, SDS-PAGE etc). Provide sufficient detail about procedures and data so that the same procedures could be exactly repeated. Moreover, the subtitles for subsection 4.1 and 4.2 are identical. Please correct.
The results are presented in detail, but no statistical processing method was used. The authors should process statistical data through key methods of interpretation and analysis in order to be able to make a critical discussion of the results. Without statistical interpretation, the study is incomplete. It looks rather than a preliminary study.
Figure 4. -I was not able to find figure 4b, even if it is presented in the figure caption.
Author Response
Rapid and easy high-molecular-weight glutenin subunit identification system by Lab-on-a-chip in Wheat (Triticum aestivum L.)
Manuscript ID: plants-983586R1
Point by point reply to comments of reviewer
We are thankful to the editorial team and anonymous reviewer for their time dedicated to this manuscript. We sincerely appreciate the comments, which helped us improved significantly the content and the quality of the manuscript. We would like to specify that all changes in the manuscript were highlighted green. We hope that the manuscript in the present form will be suitable for publication in the journal.
|
Reviewer 2
|
The paper “Rapid and easy high-molecular-weight glutenin subunit identification system by Lab-on-a-chip in Triticum aestivum” was focused on establishing a numbering system of HMW-GS subunits for Lab-on-a-chip to easily identify the HMW-GS in a short time. The study was targeted on 3, 6, and 3 of allelic variations at Glu-A1, Glu-B1, and Glu-D1 loci, which are commonly used in the wheat breeding program. The research topic is relevant because it is necessary to have rapid information about the main wheat protein due to its crucial effect on dough rheology and breadmaking. Such analysis could be complementary not only for providing information about low molecular weight glutenin subunits likely, but also to reflect on the contributions of growth/storage conditions on the quality of wheat and flour. |
|
It is expected that the application of such systems will become routine in the near future The introduction provides sufficient background and includes all relevant references. |
We are thankful to the Reviewer for his valuable comments and suggestions to improve the manuscript. |
|
Line 45- reference need to be quoted
|
We have include a reference on line 45 [1] |
|
Line 87-88- the authors can list some of the molecular markers for an easier understanding |
Lines 92-93: we have inserted the following statement: “Useful molecular markers, such as Kompetitive Allele-Specific PCR (KASP) and Single Nucleotide Polymorphism (SNP) markers,” |
|
Regarding the Materials and Methods section, it needs to be reorganized in an easier-to-follow manner (e.g samples, protein extraction, Lab-on-a-chip, SDS-PAGE etc). Provide sufficient detail about procedures and data so that the same procedures could be exactly repeated.
|
Lines 388-404: we have included a brief description of the protein separation using SDS-PAGE as follows: “4.2. Glutenin Proteins Extraction and HMW-GS Composition Identification of Genetic Resources by SDS-PAGE Glutenin subunits were also separated by sodium dodecyl sulfate-polyacrylamide gel electrophoresis (SDS-PAGE) following the procedure described earlier [45]. Briefly, the separation gel contained 1.5 M Tris-HCl (pH 8.8) and 0.27% SDS. Gels were made of 7.5% (w/v) acrylamide and 0.2% (w/v) bis-acrylamide. The stacking gel was made of 0.25 M Tris-HCl (pH 6.8), 0.2% SDS and 7.5% (w/v) acrylamide and 0.2% (w/v) bis-acrylamide. Wheat flour was suspended in 300 mL 0.25M Tris-HCl buffer (pH 6.8), containing 2% (w/v) SDS, 10% (v/v) glycerol, 5% 2-mercaptoethanol, followed by shaking for 2 h at room temperature. Then, the slurry was heated for 3 min at 95 °Cm and the supernatant was subjected to SDS-PAGE. The HMW numbering system of glutenin subunit bands and that for allelic classification at different loci previously proposed by Payne and his colleague [11] were used in the current study. For the determination of the electrophoretic mobility of each HMW glutenin subunit by SDS-PAGE, standard wheat varieties that included the spectra of the subunits expected were used. Thus, the overall quality scores of HMW glutenin subunits for a particular variety could be obtained as the sum of the scores of each individual subunit, and compared with the standard bread-making quality of the wheat varieties [46]. ” |
|
Moreover, the subtitles for subsection 4.1 and 4.2 are identical. Please correct.
|
Lines 388-404: we have replaced the subtitle with an appropriate one as 4.2. Glutenin Proteins Extraction and HMW-GS Composition Identification of Genetic Resources by SDS-PAGE Glutenin subunits were also separated by sodium dodecyl sulfate-polyacrylamide gel electrophoresis (SDS-PAGE) following the procedure described earlier [45]. Briefly, the separation gel contained 1.5 M Tris-HCl (pH 8.8) and 0.27% SDS. Gels were made of 7.5% (w/v) acrylamide and 0.2% (w/v) bis-acrylamide. The stacking gel was made of 0.25 M Tris-HCl (pH 6.8), 0.2% SDS and 7.5% (w/v) acrylamide and 0.2% (w/v) bis-acrylamide. Wheat flour was suspended in 300 mL 0.25M Tris-HCl buffer (pH 6.8), containing 2% (w/v) SDS, 10% (v/v) glycerol, 5% 2-mercaptoethanol, followed by shaking for 2 h at room temperature. Then, the slurry was heated for 3 min at 95 °Cm and the supernatant was subjected to SDS-PAGE. The HMW numbering system of glutenin subunit bands and that for allelic classification at different loci previously proposed by Payne and his colleague [11] were used in the current study. For the determination of the electrophoretic mobility of each HMW glutenin subunit by SDS-PAGE, standard wheat varieties that included the spectra of the subunits expected were used. Thus, the overall quality scores of HMW glutenin subunits for a particular variety could be obtained as the sum of the scores of each individual subunit, and compared with the standard bread-making quality of the wheat varieties [46]. |
|
The results are presented in detail, but no statistical processing method was used. The authors should process statistical data through key methods of interpretation and analysis in order to be able to make a critical discussion of the results. Without statistical interpretation, the study is incomplete. It looks rather than a preliminary study. |
Authors thank the Reviewer for the concern. We would like to specify that data of the protein amount obtained electropherogram were compared for their statistical significance difference as shown in lines 391-393 “Samples were collected in triplicate for each wheat variety use in the study. The values of protein amount obtained from different band picks were compared using the Student t-test (P<0.05). |
|
Figure 4. -I was not able to find figure 4b, even if it is presented in the figure caption. |
We apologize for the inconvenience. Figure 4 has a unique panel. Therefore, we have removed the label a and b from the manuscript. |
- Nakamura, H.; Inazu, A.; Hirano, H. Allelic variation in high-molecular-weight glutenin subunit loci of Glu-1 in Japanese common wheats. Euphtica 1999, 106, 131-138.
- Nakamura, H.; Sasaki, H.; Hirano, H.; Yamashita, A. A high molecular weight subunit of wheat glutenin seed storage protein correlates with its flour quality. Japan J. Breed. 1990, 40, 485-494.
- Payne, P.I.; Lawrence, G.J. Catalogue of alleles for the complex gene loci, Glu-A1, Glu-B1, and Glu-D1 which code for high-molecular-weight subunits of glutenin in hexaploid wheat. Cereal Res. Commun. 1983, 29-35.
- He, Z.H.; Liu, L.; Xia, X.C.; Liu, J.J.; Peña, R.J. Composition of HMW and LMW glutenin subunits and their effects on dough properties, pan bread, and noodle quality of Chinese bread wheats. Cereal Chem. 2005, 82, 345–350.

Round 2
Reviewer 1 Report
Dear authors;
thank you for taking my suggestions into account.
with kind regards
Reviewer 2 Report
I agree with this form of the manuscript. For future articles, the authors must take into consideration that more extensive statistical analysis is necessary for a better results discussion.